# Neutrophil Extracellular Traps in Atherosclerosis: Research Progress

**DOI:** 10.3390/ijms26052336

**Published:** 2025-03-06

**Authors:** Zhonghong Shi, Sihe Gong, Yanni Li, Kaijie Yan, Yimin Bao, Ke Ning

**Affiliations:** 1School of Integrative Medicine, Shanghai University of Traditional Chinese Medicine, No. 1200 Cailun Road, Shanghai 201203, China; instable_benzene@hotmail.com (Z.S.); gsh15536576127@163.com (S.G.); annie0314@163.com (Y.L.); ykj0917@126.com (K.Y.); 2School of Traditional Chinese Medicine, Shanghai University of Traditional Chinese Medicine, No. 1200 Cailun Road, Shanghai 201203, China; 3Division of Cardiovascular Medicine, Department of Medicine, Vanderbilt University Medical Center, 2220 Pierce Ave, Preston Research Building, Room 359, Nashville, TN 37232, USA

**Keywords:** atherosclerosis, neutrophil extracellular trap, inflammasomes, mitochondrial DNA, NETosis

## Abstract

Atherosclerosis (AS) is a disease characterised by the accumulation of atherosclerotic plaques on the inner walls of blood vessels, resulting in their narrowing. In its early stages, atherosclerosis remains asymptomatic and undetectable by conventional pathological methods. However, as the disease progresses, it can lead to a series of cardiovascular diseases, which are a leading cause of mortality among middle-aged and elderly populations worldwide. Neutrophil extracellular traps (NETs) are composed of chromatin and granular proteins released by neutrophils. Upon activation by external stimuli, neutrophils undergo a series of reactions, resulting in the release of NETs and subsequent cell death, a process termed NETosis. Research has demonstrated that NETosis is a means by which neutrophils contribute to immune responses. However, studies on neutrophil extracellular traps have identified NETs as the primary cause of various inflammation-induced diseases, including cystic fibrosis, systemic lupus erythematosus, and rheumatoid arthritis. Consequently, the present review will concentrate on the impact of neutrophil extracellular traps on atherosclerosis formation, analysing it from a molecular biology perspective. This will involve a systematic dissection of their proteomic components and signal pathways.

## 1. Introduction

NETs are extracellular structures composed of chromatin and granular proteins that are used to capture and kill microorganisms. When stimulated, neutrophils undergo a series of changes, including loss of nuclear shape, disintegration of the nuclear and cytoplasmic granular membranes, and rupture of the cell membrane. This process of cell death, which differs from both apoptosis and necrosis, is known as NETosis [1]. The release of NETs has been demonstrated to be an effective mechanism in immobilising pathogens, preventing their propagation and creating spatial opportunities for elevated local concentrations of antimicrobial factors. However, if NETs are not fully eradicated or exhibit atypical increases, they have the potential to promote the pathogenesis of numerous inflammatory diseases, including atherosclerosis. Accumulating evidence suggests that NETs also play a significant role in autoimmune diseases, such as systemic lupus erythematosus and rheumatoid arthritis, where they contribute to disease progression by inducing autoimmunity and amplifying chronic inflammation [2].

NETosis is a distinctive form of immune response and a unique form of cell death that differs from both necrosis and apoptosis [3]. Neutrophils release NETs to protect the body. This process involves the rupture of the neutrophil nuclear membrane, which forms a fibrous network composed of antimicrobial proteins such as myeloperoxidase (MPO) and neutrophil elastase (NE), as well as chromatin. The purpose of NETosis is to capture and degrade pathogens.

The process of NETosis is primarily triggered by oxLDL and cholesterol stimulation, leading to the release of mtDNA and the accumulation of reactive oxygen species (ROS) in neutrophils. This, in turn, results in histone phosphorylation and the depolymerisation of chromatin, ultimately causing nuclear membrane damage [4].

In order to facilitate the process of segmenting the development of NETs, the Results section divides the development of NETs into three stages: the pre-NET release stage, the NET release stage, and the post-NET release stage. The pre-NET stage will focus on the factors that lead to the occurrence of NETosis, as well as some of the target factors involved in the process, and explain their role in the development of atherosclerosis. The NET release stage refers to the physiological phenomenon of NETosis, and the text will discuss in detail the dynamic changes inside and outside neutrophils during this process. The post-NET release stage focuses on analysing the impact of the composition and physical and chemical structure of NETs on the development of atherosclerosis.

It is evident that NETs, as a distinctive immune response and a novel form of cell death, are intricately linked to AS in terms of their activation, production, and release. This article categorises the cellular process of NETs into three distinct stages: pre-NET stage, NETosis, and NET release stage. It delves into the pertinent mechanisms of NETs and their function in the progression of atherosclerosis. The objective is to analyse the role of NETs in different stages of AS from a molecular biology perspective and to summarise treatment methods, providing a new perspective for the treatment and prevention of cardiovascular diseases such as atherosclerosis.

## 2. Pre-NET Stage: Factors Associated with the Development of NETosis

NETs can be produced in the arteries through a variety of mechanisms, and the process can be activated by a variety of antigens, including various bacteria, fungi, viruses, inflammatory factors, immune complexes, and other stimuli [2,5]. These antigens have been demonstrated to induce the production of NETosis and the formation of NETs through various mechanisms, including the peptidylarginine deiminase 4(PAD4)-dependent pathway [6] and the cell cycle-dependent pathway [7]. Among these, the formation and accumulation of ROS play a pivotal role in the production of NETs [8]. The content of oxidised low-density lipoprotein (oxLDL) and cholesterol has been demonstrated to play a pivotal role in the formation of ROS. Concurrently, oxLDL and cholesterol have been observed to stimulate monocytes and activated macrophages to form ROS, promote their development into foam cells, and lead to the release of pro-inflammatory cytokines and enhanced phagocytic activity, which can lead to the development of atherosclerosis [9,10].

The formation of ROS is a complex process involving the participation of multiple cytokines and signal pathways [11]. The ensuing discourse in this review will chiefly encompass cytokines (e.g., interleukin, TNF-α, and IFN-γ) and signal pathways (e.g., NF-κB and cGAS-STING) and will elucidate their function in the genesis of NETs. Concurrently, these NET-promoting factors are also implicated in the development of atherosclerosis, a subject that will also be addressed herein.

### 2.1. Cytokiness

Cytokines, defined as a class of proteins that regulate bodily functions, have been demonstrated to play a pivotal role in the modulation of inflammatory and immune responses [12,13,14]. These cytokines have been found to be implicated in the development of atherosclerosis and NETs [15,16].

It has been demonstrated through experimentation that pro-inflammatory cytokines (e.g., IL-8 and IL-18) play a pivotal role in the process of inducing neutrophils to produce NETs [17]. Furthermore, experimental evidence has demonstrated that IL-8 and IL-1β have the capacity to induce elastase release in a concentration-dependent manner, which subsequently results in NETosis. Moreover, TNF-α, IL-8, and IL-1β have been observed to enhance the production of NETs [18].

Among them, IL-8 is a chemokine. When the body is stimulated by inflammation, such as infection or cell damage, chemokines are released from injured tissues or inflammation sites to attract and guide immune cells to the target site. This process is known as neutrophil recruitment and colonisation [19]. Neutrophils have been identified as the causative agents of transient interactions between white blood cells and endothelial cells in atherosclerotic lesions (e.g., leukocyte infiltration in atherosclerosis [20]) [21,22]. In sterile inflammation induced by the IL-1 family, serine proteases and nuclear bodies produced by neutrophils have been shown to lead to thrombosis [23].

The IL-1 family of proteins comprises a multitude of subtypes, including IL-1, IL-18, IL-33, and IL-36. These proteins have the capacity to activate pro-inflammatory pathways by inducing the expression of other cytokines (e.g., IL-6 [24] and IL-17 [25]) [26,27]. The resultant events may include vascular endothelial damage, oxidative stress, foam cell formation, and accelerated atherosclerotic plaque formation [28,29,30,31,32,33,34,35,36,37,38]. The IL-36 pathway has been shown to stimulate the activation of the NLRP3 inflammasome [39].

IL-18, a constituent of the IL-1 family, has been demonstrated to activate the NF-κB signalling pathway, consequently resulting in an augmentation in the expression of downstream inflammatory mediators [24], including IFN-γ. IFN-γ has been observed to elicit a series of reactions that are conducive to the development of atherosclerosis, such as endothelial damage and the upregulation of inflammatory factor secretion [31,32].

### 2.2. NF-κB Signalling Pathway

The NF-κB signalling pathway is a pivotal regulator of inflammatory responses and has a significant role in atherosclerosis and the formation of NETs [40]. The NF-κB family comprises five distinct members, which can form different dimers [41]. NF-κB has been demonstrated to promote the activation of neutrophils and the production of NETs by activating genes associated with NET formation, including inflammatory mediators and chemokines [19].

A significant body of experimental evidence has demonstrated a reciprocal relationship between the occurrence and release of NETs and NF-κB [42]. NF-κB is activated by an IgG fragment, which in turn promotes the production of ROS, thereby enhancing the formation of NETs [43]. Inhibiting NF-κB activation has been demonstrated to result in a decrease in IL-8 expression and a decrease in NET production. This suggests that NF-κB plays a pivotal role in the formation of NETs [44,45].

A plethora of factors associated with atherosclerosis, including haemodynamics, cytokines, bacterial and viral infections, and lipid peroxidation, have been demonstrated to activate NF-κB [46]. NF-κB also plays a regulatory role in the expression of genes associated with cytokines (e.g., TNF, IL-1, IL-6, IL-8) [42], adhesion molecules (e.g., VCAM-1, ICAM-1, P- and E-selectin) and chemokines (e.g., MCP-1), all of which are implicated in the development of atherosclerosis [47]. Activation of NF-κB has been identified in human atherosclerotic plaques, with a greater prevalence observed in acute coronary syndromes associated with atherosclerosis. The nuclear translocation of the NF-κB subunit RelA has been detected in smooth-muscle cells, macrophages, and endothelial cells in atherosclerotic lesions, and this phenomenon is more prevalent in unstable atherosclerotic plaques [48,49].

Inflammasomes are protein complexes that recognise a wide range of inflammation-inducing stimuli and control pro-inflammatory cytokines [50]. The activation of inflammasomes necessitates the presence of both an initiating and an activating signal. The initial signal has been shown to induce the expression of NLRP3 and the IL-1β precursor by activating the NF-κB pathway. The signal provided by the NF-κB activating factor is necessary for the activation of NLRP3 [39].

### 2.3. cGAS-STING Signalling Pathway and Mitochondrial DNA (mtDNA)

The cGAS-STING signalling pathway represents an intracellular immune sensing mechanism that is activated in response to the accumulation of abnormally accumulated intracellular DNA, a phenomenon that occurs as a consequence of pathological conditions such as infection and intracellular damage. Consequently, this pathway plays an essential role in the body’s immune response.

As demonstrated by previous experiments, mtDNA has been observed to activate the cGAS-STING signalling pathway [51], resulting in an elevated expression of NF-κB activation proteins and the initiation of an inflammatory response. This inflammatory response, in turn, promotes the activation of neutrophils and the process of NETosis [52]. The release of mitochondrial endogenous DNA has been shown to activate the cGAS-STING signalling pathway, triggering a series of subsequent reactions. These findings are consistent with the response process of cytoplasmic mitochondrial DNA [52]. Furthermore, STING has been demonstrated to promote NET formation via the ROS-ERK signal [53].

The process of GSDMD cleavage in neutrophils is initiated by the action of inflammasomes [54], leading to the self-destruction of the inflammasomes and the subsequent release of IL-1β. Experimental evidence has demonstrated the capacity of GSDMD-n fragments to contribute to the activation of the cGAS-STING pathway. It has been demonstrated that GSDMD can cause mtDNA to leak into the cytoplasm of neutrophils, thereby activating the cGAS-STING signalling pathway [51,55].

In the process of promoting atherosclerosis, the cGAS-STING pathway has been shown to trigger DNA damage, promote the expression of pro-inflammatory cytokines (such as IL-6) [56], activate macrophages [52], and induce the pyroptosis of vascular endothelial cells [57], as well as other cellular changes, thus promoting atherosclerosis. In atherosclerotic mice, the expression of STING and the stimulants of DNA damage increases [58]. The experimental observations outlined here demonstrate the pivotal role of the cGAS-STING pathway in the development of atherosclerosis.

Figure 1 summarizes the processes involved in the Pre-NET Stage systematically, including cytokines, signaling pathways in the formation of NETs and the occurrence and development of atherosclerosis.

## 3. NET Release Phase: The Mechanism of NETosis

Neutrophils are sufficiently stimulated by exogenous antigens (e.g., bacteria or viruses) or endogenous antigens (e.g., cytokines). In addition, surface antigens on thrombocytes (e.g., FcγRII [59]) and endothelial cells (e.g., VCAM-1, AECA [60]) can also play a crucial role in the activation of neutrophils, further contributing to the inflammatory response. These stimuli can induce neutrophils to actively release NETs, a process referred to as NETosis.

NETosis is categorised into two distinct classifications: suicidal NETosis and non-suicidal NETosis. The suicidal NETosis pathway is characterised by a duration of several hours and is contingent upon stimuli such as PMA (phorbol-12-myristate-13-acetate) and the subsequent activation of NADPH oxidase. Additionally, the deconcentration of chromatin is a prerequisite for this process, and this is mediated by MPO and NE. The consequence of these processes is the production of a mixture of DNA and granular proteins, which are extruded from the plasma membrane through pores. The vital NETosis pathway has been demonstrated to maintain the ability of neutrophils within 30 min and during NETosis [61]. At present, the majority of experiments on NETs involve the suicidal NETosis process. The NETosis process comprises four distinct stages: mitochondrial damage; disruption of the cytoskeleton and nuclear envelope; chromatin depolymerisation; and assembly of antimicrobial proteins on the chromatin scaffold [62].

### 3.1. Mitochondrial NADPH Oxidase Process

The process of neutrophil extracellular trap formation is dependent on the activation of the Raf-MEK-ERK pathway. The accumulation of large antigenic loads in proximity to neutrophils has been observed to trigger the activation of the neutrophil MEK-ERK pathway, thereby initiating the production of the downstream protein kinase C (PKC). PKC has been demonstrated to act as a second messenger, thereby activating the mitochondrial NADPH oxidase mechanism. This, in turn, has been shown to produce a significant amount of ROS. These large amounts of ROS have been identified as playing a key role in the initiation of NETosis [63]. Concurrently, calcium ion channels are activated, calcium ions enter the mitochondria, and a substantial amount of ROS is produced.

The release of NE from the azurophilic granule membrane is promoted by hydrogen peroxide in ROS, which occurs from the release of calcium ions into the mitochondria. MPO plays an important role in the localization of hydrogen peroxide to the complex [64]. At this time, MPO remains in the azure bodies of azure granules. The action of MPO results in the release of a significant quantity of NE from the azure granules.

Concurrently, following the generation of substantial levels of ROS, neutrophils trigger the release of a considerable quantity of mtDNA within the cGAS-STING signalling pathway [55,65].

### 3.2. Disintegration of the Cytoskeleton and Nuclear Membrane

Following activation, NE binds to F-actin, resulting in the dissociation of actin and subsequent inhibition of actin dynamics.

Rupture of the nuclear membrane is a pivotal step in the process of NETosis which determines whether the process of NETosis is reversible. During this phase, the aberrant activation of cyclin-dependent kinase 4/6 (CDK4/6) accelerates the process, causing the phosphorylation of nuclear envelope proteins and further leading to the destabilisation of the nuclear envelope structure. Furthermore, the mechanism of CDK4/6 associated with microfilaments has the potential to initiate mitosis and promote the disintegration of the nuclear envelope [4]. Furthermore, although experiments have demonstrated that MPO enters the nucleus concomitantly with NE, its function in chromatin disassembly is independent of its enzymatic activity, and its specific mechanism of action remains to be studied [4].

About 1 h after nuclear membrane breakdown and chromosome decondensation occurs, the neutrophils have secreted the enzymes required for the ensuing processes: histone citrullination and laminin phosphorylation. These processes prepare the cells for the subsequent morphological changes that occur in the late stage. Until the termination of this phase, the NETosis process is primarily governed by enzymatic activity. Following a duration of 80 min, the nuclei undergo a transformation, losing their lobular structure and instead becoming enlarged and occupying the majority of the intracellular space. Concurrently, the cells maintain their level of activity [66].

At this stage, chromatin undergoes a rapid swelling, resulting in its exposure to the cell membrane. At this time, due to the counterforce that maintains the DNA/chromatin structure being less than the entropy pressure, the chromosomes gradually expand. However, the restructuring of F-actin in the cytoskeleton, the disorganisation of actin and the microtubule apparatus, leads to damage to the mechanical properties of the cell, which facilitates deformation and rupture of the cell membrane. During this process, MPO converts hydrogen peroxide into hypochlorous acid, which activates NE. The activation of NE subsequently activates the downstream pore-forming protein (GSDMD), resulting in the formation of pores between the nuclear and plasma membranes. This process leads to the destruction of the original structure of the cytoskeleton, a reduction in membrane stability, and the facilitation of changes in cell shape and the release of DNA [67,68]. At this stage, the activity of enzymes becomes a minor factor affecting the NETosis process, and the process of NETosis is irreversible at this point. There is no significant decrease in ATP levels in the cells at this stage, and ATP is not consumed at this stage.

### 3.3. Chromatin Decondensation

The process of chromatin depolymerisation is initiated by the activation of peptidyl arginine deiminase 4 (PAD4) [19]. The enzyme in question catalyses the guanidinylation modification of the arginine residues of histones H3 and H4. This results in a significant reduction in the positive charge density of the histones. The consequence of this is a weakening of the electrostatic interactions between the DNA in the chromatin. This, in turn, subsequently triggers the initial depolymerisation of the chromatin [68]. Concurrently, NE outside the nucleus translocates into the nucleus, where it plays a pivotal role in two distinct ways. Primarily, NE degrades histone H1, thereby disrupting the spatial structure of the chromosome. Moreover, NE also degrades core histone H4, which further promotes complete depolymerisation of the chromosome. These two processes exhibit a discernible temporal sequence, and the continuous modification and degradation of histones disrupts the ionic balance in the DNA–histone complex. Concurrently, the entropic pressure induced by the substantial intrinsic radius of the human genome (150-200 microns) prompts a further expansion of the chromosomes, a process that is irreversible [69]. Concurrently, the intracellular ATP level drops sharply (by about 70%) [69]. The imbalance in energy metabolism has been shown to exacerbate this process by inhibiting ATP-dependent chromatin remodelling enzymes (such as topoisomerases).

### 3.4. Release of NET Contents

About 220 min after the NETosis process begins, the cell membrane undergoes a rupture, resulting in the release of NETs contents. The composition of NETs is dominated by nuclear and granular contents, with nucleic acids and proteins constituting the predominant components. The core proteins are histones, NE, and MPO. Experimental evidence has demonstrated that NETs are not detected until this stage, and that the rupture of the cell membrane is a necessary step for the release of NETs [25]. The DNA in NETs is predominantly derived from the nucleus and mitochondria of neutrophils. The reticular structure of NETs has the capacity to capture and digest microorganisms, and the protease cell degradation factors contained within them can also play an anti-inflammatory role [66].

Figure 2 summarizes the schematic diagram of the molecular mechanism of neutrophil suicidal NETosis, including four steps: mitochondria activation, rupture of the nuclear membrane and disassembly of the cytoskeleton, chromatin decondensation and NET release.

## 4. Post-NET Release Phase: Effects of NET Release on Atherosclerosis

Atherosclerosis is now widely regarded as a disease arising from an accumulation of lipids and subsequent inflammation [48]. The initiating factor for atherosclerosis is typically vascular damage. The progression of atherosclerosis is typified by three significant stages: the formation of fatty streaks, the induction of atherosclerosis, and the development of atherosclerotic plaques [70].

The NETosis process and the AS process are in a mutually reinforcing relationship. The NETosis process is involved in the processes of AS-related factors such as hyperlipidaemia and oxidative stress, thereby affecting the occurrence of AS, the formation of AS plaques, and the unstable rupture of AS plaques. The main components of NETs are nuclear chromatin, serine proteases, and MPO. Of these, MPO and NE have been shown to be important components involved in the immune response [51]. Furthermore, the presence of additional antigens, such as double-stranded DNA and histones, within NETs, serves to further reinforce the immune response [2].

### 4.1. MPO and Oxidative Stress

MPO in NETs has been observed to bind to the mannose receptor CD206 on the surface of macrophages. This binding has been shown to induce the production of ROS, oxidise LDL to oxLDL through its catalytic activity, and cause macrophages to release cytokines, mainly IL-1β [71]. The subsequent uptake of ox-LDL molecules by macrophages is facilitated via the scavenger receptor-A family, resulting in the formation of foam cells laden with lipids [25]. The accumulation of yellow foam cells on the arterial wall leads to the formation of fatty streaks. The role of hypochlorous acid produced by MPO in immunity is well documented, but activated MPO can also cause damage to surrounding tissues, which in turn triggers an inflammatory response [72]. Furthermore, MPO-DNA complexes have been shown to activate neutrophils, thereby maintaining the development of inflammation [73]. Furthermore, MPO-DNA complexes serve as the primary markers of NETs, in addition to their role in stimulating the production of NETs [74]. The MPO-DNA complex activates Toll-like receptor 9 (TLR9) and autoreactive B lymphocytes by binding to Fcγ receptors (FcγR) on plasmacytoid dendritic cells (pDC). This process leads to a strong IFN-1 response, which promotes the development of atherosclerosis by triggering inflammation, activating endothelial cells to secrete chemokines, inducing vascular wall remodelling, and promoting foam cell formation. The body’s long-term accumulation of IFN-1 has been demonstrated to be a contributing factor to the development of atherosclerosis [75,76].

### 4.2. NE and Thrombosis

NE from NETs has been shown to degrade tissue factor pathway inhibitor (TFPI), antithrombin, and activated protein C [23]. In addition, exogenous NE may originate from necrotic cells in the immediate vicinity of the cell. Consequently, NETs promote the expression of von Willebrand factor (VWF) and p-selectin on the surface of vascular endothelial cells, thereby capturing platelets and red blood cells to form a fibrin deposition scaffold [77]. In advanced stages of AS, the proteolytic proteins derived from NETs (e.g., NE) and the proteolytic enzymes secreted by macrophages and T lymphocytes on AS plaque (e.g., matrix metalloproteinases) can destroy the structure of AS plaque, leading to leakage of the extracellular matrix outside the plaque and causing thrombosis [2].

### 4.3. Cytotoxic and Prothrombotic Effects of Histones and Other Neutrophil-Derived Protein

Neutrophil-derived proteins that are part of the NET skeleton (e.g., cathepsin G or cathelicidin-related antimicrobial peptide (CRAMP, cathelicidin-related antimicrobial peptide)) can be deposited and transported across the inflamed endothelial surface of atherosclerotic blood vessels [78]. These proteins have been demonstrated to exert a chemotactic effect on monocytes [79,80]. It has been demonstrated that protease G can enhance tissue factor- and factor XII-driven coagulation via proteolytic activation of tissue factor by NE via tissue factor pathway inhibitor (TFPI) [23]. Furthermore, platelets activated by protease G have been observed to be able to be activated by TLR4 and subsequently bind to endothelial-adherent neutrophils, which in turn produce NETs [23]. It has been demonstrated that thrombocytes activated by tissue plasminogen activator can be stimulated by TLR4 and bind to endothelial adherent neutrophils, thereby inducing NETosis [81].

NETs have been shown to contain histones, which have been demonstrated to be capable of causing bystander injury. Extracellular histones have been shown to activate TLRs, leading to the production of thrombin, platelet activation, microaggregation, and thrombocytopenia. Furthermore, histones have been identified as potent cytotoxic molecules for endothelial cells [4]. Histone H3 and H4 have been observed to aggregate with platelets, induce calcium influx, recruit plasma adhesion proteins (e.g., fibrinogen) to induce platelet aggregation, and promote the formation of microthrombi [23,82]. During this process, the activation of endothelial cells will result in the release of inflammatory factors, including IL-8 and IL-1β. These, in turn, will further promote the activation of neutrophils and the formation of NETs. The released NETs have the capacity to induce the local activation of the alternative complement cascade in proximity to vascular endothelial cells (dependent on the formation of anaphylatoxins C3a and C5a and membrane attack complexes) [83]. This, in turn, promotes the sustained activation, detachment and eventual erosion of endothelial cells [84]. In vitro, intact NETs have been demonstrated to be incapable of initiating coagulation; however, some of the DNA isolated from them has been observed to possess procoagulant activity. This phenomenon may be attributed to the negative charge carried by the DNA, which has the capacity to modify the structure of the protein [85]. The addition of histones in an artificial manner has been observed to result in the binding of histones to the DNA, forming a substance analogous to chromatin. This process has been shown to result in the loss of the coagulative effect [86]. This derived DNA has been observed to bind to the elevated levels of antimicrobial peptides present within atherosclerotic lesions, thereby triggering a response in plasmacytoid dendritic cells (pDCs). This response, in turn, drives the production of anti-double-stranded DNA antibodies, which have been associated with the development of early atherosclerotic lesions [87]. The DNA, which is present in high concentrations within the structure, has been observed to bind to antimicrobial peptides (e.g., defensins and the antimicrobial peptide LL-37) in atherosclerotic plaques. This binding has been shown to trigger a response in plasmacytoid dendritic cells (pDCs). The resultant response has been found to lead to the production of anti-double-stranded DNA antibodies, which have been associated with the development of atherosclerosis [87].

### 4.4. The Inflammatory Effect of mtDNA

mtDNA, as part of the NET skeleton, is a danger signal-associated molecular pattern (DAMP) in innate immunity. The distinguishing feature of mtDNA is its methylation pattern, which differs from that of nuclear DNA. This distinction enables direct recognition by the immune system, leading to an inflammatory response [88,89]. This response results in mtDNA damage and the activation of the associated inflammasome, NLRP3, which in turn triggers a series of subsequent reactions. It has been demonstrated by preceding experiments that the process of atherosclerosis is associated with damage to mtDNA [90,91]. Furthermore, it has been established that mtDNA damage occurs prior to the onset of atherosclerosis. The precise mechanism by which mtDNA exacerbates atherosclerosis remains to be elucidated [92]. However, it may be associated with an augmented inflammatory response and accumulation of monocytes concomitant with mtDNA [90].

### 4.5. The Immune Metabolic Process in Relation to Non-Coding microRNAs (miRNAs)

miRNAs have also been identified in NETs, where they are found mixed in with DNA. These NET-miRs interact with extracellular miRNAs to regulate the production of TNF-α by macrophages, which in turn triggers an inflammatory response. One of these miRNAs, miR-142-3p, is a functional form of RNA. The transfer of these miRNAs to neighbouring cells during NET interactions results in the regulation of the expression of their mRNA targets. miR-142-3p functions as a regulatory factor by downregulating PKCα (the isoform α of the Protein Kinase C, PKC), a key player in the regulation of macrophage function following the activation by lipopolysaccharide (LPS). PKCα is involved in the signal pathway leading to the production of pro-inflammatory mediators such as IL-6 and TNF-α [93]. It has been demonstrated that NET components DNA and H4 can lead to increased expression of HNF4A mRNA, which may indicate that they are partially involved in the regulation of coagulation factors. Furthermore, H4 has been shown to induce the expression of specific microRNAs (miRNAs) in the miR-17/92 cluster, which partly explains the induction of tissue factor (TF) expression by histone H4 [94].

### 4.6. Inflammatory Bodies: The Cross-Regulatory Hub of NETs and AS

Inflammasomes, which are mentioned numerous times in the formation of NETs, not only play an important role in the process of NETosis but also play an important role in the formation of atherosclerosis [95]. Inflammasomes are considered to be the link between the inflammatory response and lipid metabolism, because two components abundant in atherosclerotic plaques, crystalline cholesterol and oxLDL, activate NLRP3 inflammasomes. Furthermore, oxidative stress, mitochondrial dysfunction, endoplasmic reticulum stress, and lysosomal rupture associated with the activation of inflammatory bodies are also considered to be important events in atherosclerosis [24,96]. NLRP3 inflammasomes have been demonstrated to activate the scavenger receptor CD36 on the surface of macrophages in the presence of NETs, thereby stimulating macrophages to synthesise the pro-inflammatory factor IL-1β [25]. It has been established that this process also activates helper T cells (Th17), which in turn release IL-17 and promote the recruitment of immune cells in atherosclerotic plaques. Atherosclerotic blood flow has been shown to induce NLRP3 inflammasomes in endothelial cells by activating SREBP2 [25]. The increased innate immunity of the endothelium has been demonstrated to act synergistically with hyperlipidaemia to cause the topographic distribution of atherosclerotic lesions [97].

### 4.7. DNA–Histone Complex

NETs and the antimicrobial proteins they enclose (antimicrobial proteins, e.g., NE, MPO, IFN-γ, LL-37, are a class of proteins secreted or stored by host cells that have the function of directly or indirectly inhibiting or killing pathogenic microorganisms) have been shown to activate inflammatory bodies. These, in turn, have been demonstrated to prime inflammatory cytokines, further stimulating surrounding neutrophils to undergo NETosis (NETosis is the process by which neutrophils release an extracellular web of DNA, or NET, which can entrap bacteria). This positive feedback loop represents a significant mechanism in the pathogenesis of various diseases [86].

The DNA–histone complex present in the components of NETs has been shown to form a reticular fibrous structure. Furthermore, the proteases (e.g., MPO and NE) contained within this complex bind to specific antigens on the surface of the vascular wall or blood cells. This, in turn, has been demonstrated to trigger a series of reactions that lead to the development of AS [98]. Histones H3/H4 in NETs have been observed to bind to negatively charged phosphatidylserine (PS) on the platelet surface through electrostatic forces, thereby activating its surface receptors glycoprotein VI (GPVI) and Toll-like receptors (TLR4/9). This process has been shown to promote platelet activation and aggregation [99]. The presence of extracellular DNA in NETs has been observed to facilitate binding to the adhesion molecule P-selectin on the surface of platelets. This interaction enables the scaffold structure of NETs to become firmly attached to the vascular endothelium, thereby contributing to the process of plaque formation. Furthermore, the presence of histones within NETs has been shown to recognise blood group antigens on the red blood cell membrane. This recognition process leads to the destruction of the red blood cell membrane, thereby accelerating the process of thrombosis formation [100]. The binding of histones to heparan sulphate proteoglycans (HSPGs) and integrin αvβ3 on the surface of vascular endothelial cells has been shown to trigger a local inflammatory response [101]. Experimental evidence has demonstrated that, subsequent to the degradation of the DNA backbone of NETs using DNase, the histone–HSPG complex and NE-PAR1 complex persistently adhere to the endothelium for an extended duration, thereby activating complement (e.g., C5a), inducing substantial levels of ROS, and inflicting harm upon adjacent tissues [102]. The mechanisms by which these targets interact with antigens mutually promote the development of AS from various perspectives, including the pro-inflammatory response, endothelial damage, and plaque formation.

It is therefore evident that the interaction between procoagulant factors, NETs, and activated platelets forms a positive feedback loop, promoting the development of inflammation and the atherosclerotic process in a series of reactions, which will eventually evolve into a thrombus and trigger a series of cardiovascular diseases. The blood cells aggregated in the thrombus can promote the NETosis process through a series of mechanisms.

Figure 3 summarizes the role of NETs in AS and its related mechanisms, which includes seven part: MPO, NE, histone, miRNAs, mtDNA, inflammasomes. In addition, it explains the mutually reinforcing relationship between AS and NETosis process.

## 5. Conclusions

This review focuses on the mechanisms and effects of NETs in the development of AS. Due to the complexity of the mechanisms involved, this article divides NETs from production to release into three stages which describe the interaction between NETs and AS: pre-NET release, NETosis, and post-NET release.

The pre-NET stage is the preparatory stage for NETosis to start. This stage is mainly caused by exogenous stimuli such as oxLDL and cholesterol. Under the action of a series of cytokines (such as IL-8, IFN-γ, etc.) and key signal pathways (such as NF-κB, cGAS-STING), resulting in a series of cellular physiological phenomena such as ROS production in neutrophils, mtDNA release, nuclear membrane rupture, and chromatin depolymerisation, ultimately leading to the release of NETs. This process is called NETosis.

The main components of NETs include histones, MPO, NE, and chromatin. The mechanism by which they act on AS is diverse, but the core mechanisms include endothelial damage and immune response caused by inflammatory response, promotion of thrombosis, and plaque instability.

The vicious cycle caused by the interaction between NETs and AS after release is also a novel perspective in this mechanism. NETs can promote the development of AS through a series of cellular pathways, and AS is also a key factor in stimulating the NETosis process.

Therefore, targeted therapeutic strategies for the mechanism of NETosis have great potential in the treatment of AS. It should not be limited to inhibiting the development of AS, but should focus on new approaches from the immune system, haemodynamics, etc.

## 6. Discussion

The development of AS is characterised by a complex inflammatory response and a series of immune system regulatory mechanisms, with a growing body of research highlighting the role of NETs as a key factor in the progression of AS. This review provides a comprehensive discussion on the mechanism of NETosis and its role in the pathophysiology of AS, offering novel insights into the potential therapeutic targets for NETs in the management of AS [1].

The initiation of NETosis can be summarised as a synergistic effect of pro-inflammatory factors (e.g., IL-8 [19], IFN-γ [31,32], IL-1 family proteins [28,29,30,31,32,33,34,35,36,37,38]) and key signalling pathways (e.g., NF-κB [40], cGAS-STING [53]). During NETosis, mitochondrial oxidation leads to the release of mtDNA and the production of large amounts of ROS [8]. The production of ROS leads to the subsequent PAD4-mediated histone citrullination, which drives the process of chromatin decondensation [19]. However, the present paper proposes that the perforation of the plasma membrane mediated by GSDMD also leads to the release of mtDNA, a process that does not lead to mitochondrial damage [54]. This is related to the mechanism of another subtype of NETosis, namely non-suicidal NETosis (vital NETosis [4]).

Furthermore, the decondensation of chromatin caused by histone H3/H4 citrullination in the NETosis process is an epigenetic modification, which suggests that epigenetic regulation may become a new target for intervention in NETosis. Experiments have shown that PAD4 inhibitors can effectively inhibit the deposition of NETs in animal models [103]. However, the long-term effects on the overall AS process still need to be assessed in clinical studies.

Despite the plethora of studies that have elucidated the mechanisms involved in NETosis, numerous issues remain to be addressed. Primarily, the extant experimental findings are predominantly derived from in vitro and murine models, which may not accurately reflect the development of AS in humans and the effects of NETs. Moreover, the NETosis process persists for approximately 220 min, and there is a paucity of research on the dynamic changes in AS during this period. Finally, the feasibility and safety of therapeutic targets derived from experimental findings cannot be guaranteed.

This review of the various mechanisms by which NETs are involved in AS has resulted in the following treatment ideas being proposed: the use of enzymes such as DNaseI to decompose NETs, with the aim of reducing vascular inflammation and the formation of atherosclerotic plaques [104] and the use of inhibitors of relevant signal pathways to block key signal nodes and inhibit the pathological effects of NETosis while retaining the antibacterial function of neutrophils [105,106]. Targeting the composition of NETs, the use of related drugs (such as the use of corresponding proteases for antimicrobial proteins such as NE and MPO and the use of corresponding anti-histone antibodies for histones) can regulate the body’s immune response.

In summary, the correlation between NETosis and AS therefore gives rise to a number of new research areas in cell biology. It is clear that further research in the future, combining interdisciplinary approaches and clinical trials, is necessary. Such research could facilitate the analysis of mechanisms and thus enable precise future interventions in AS. The aim is to provide patients with AS with more effective and safer treatment options.

## Figures and Tables

**Figure 1 ijms-26-02336-f001:**
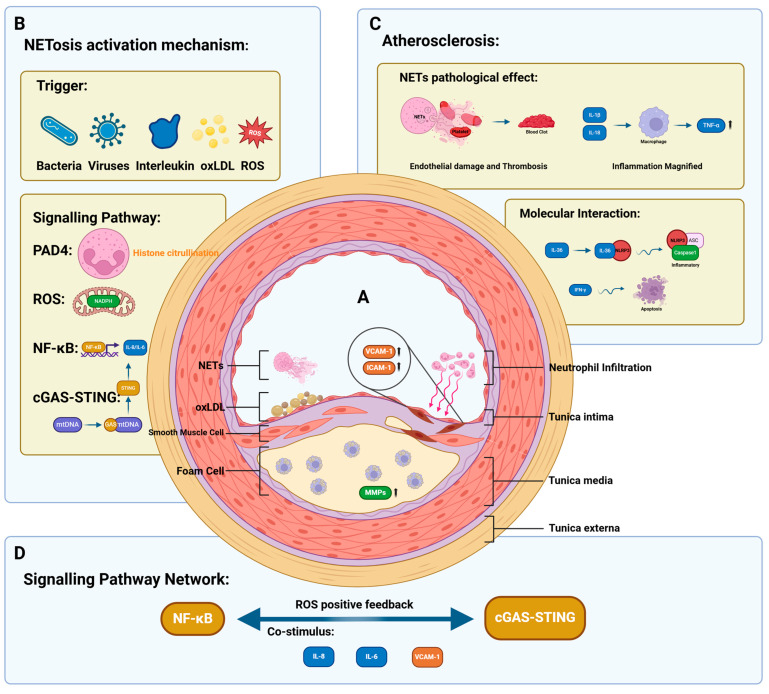
This figure shows the mechanistic interplay between NETosis and atherosclerosis development in the pre-NET stage. Black upward arrow indicates the expression of this substance is upregulated; indigo arrow indicates the substance binds to the corresponding receptor or reaches the corresponding location through inteacellular/extracellular movement; purple arrow indicates that up-regulation of gene expression leads to up-regulation of expression of specific substances; indigo wavy arrow indicates the former leads to the production of cellular biological activities in the latter through some cellular pathways. (**A**). The central part shows a cross-section of an artery and describes the situation inside the blood vessel during the development of atherosclerosis. Damaged endothelial cells upregulate the expression of chemokines such as VCAM-1/ICAM-1, which can promote neutrophil infiltration. Foam cells and oxLDL gradually accumulate in the intima, and smooth-muscle cells migrate like an intermediate layer. NETs and oxLDL adhere to the intimal surface to form the basic structure of plaque attachment. (**B**). The upper left part shows the stimuli involved in NETosis activation in the pre-NET stage. These include bacteria, viruses, some pro-inflammatory cytokines (such as IL-6, IL-8, and TNF-α), oxLDL, and ROS. There are also signalling pathways involved in this process, such as the NADPH oxidase pathway, which produces a large amount of ROS; the NF-κB signalling pathway, which upregulates the expression of IL-6/IL-8; and the cGAS-STING signalling pathway, which is the key mechanism of mtDNA leakage and NETosis. (**C**). The upper right panel shows the effect of factors that promote NETosis in the pre-NET stage on the development of atherosclerosis. For example, pro-inflammatory factors such as IL-6/IL-8 upregulate the expression of TNF-α in macrophages, amplifying the inflammatory response; IFN-γ induces apoptosis, leading to endothelial damage; and IL-36 activates inflammasomes. (**D**). The lower part shows the interaction between the signalling pathways, i.e., the common stimuli for NF-κB and cGAS-STING (e.g., IL-8/IL-6, VCAM-1), and the positive feedback between the two, leading to a massive production of ROS. Abbreviations: oxLDL (oxidised low-density lipoprotein).

**Figure 2 ijms-26-02336-f002:**
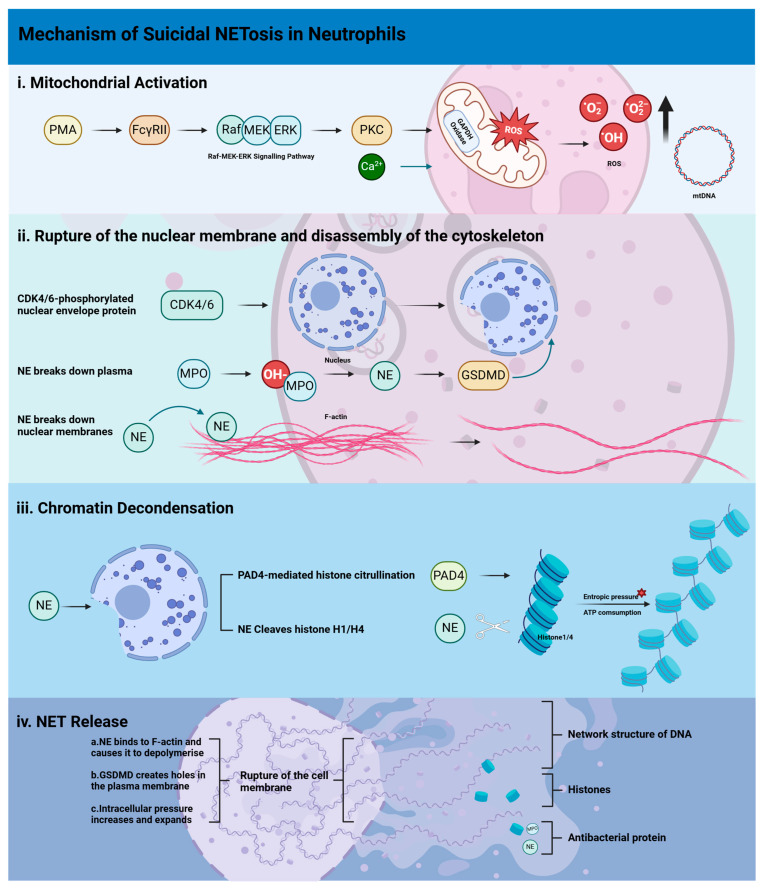
This figure shows a schematic diagram of the molecular mechanism of neutrophil suicidal NETosis. Black upward arrow indicates the expression of this substance is upregulated; indigo arrow indicates the substance binds to the corresponding receptor or reaches the corresponding location through inteacellular/extracellular movement; black arrow indicates the former promotes the upregulation of expression or cell behavior of the latter; red star symbol next to the text indicates that this influencing factor is difficult to display with symbols. The process is principally divided into four stages: i. Stimuli from PMA activate the downstream second messenger protein PKC through the Raf-MEK-ERK signalling pathway, which in turn triggers mitochondrial NADPH oxidase to produce a large amount of ROS. Simultaneously, calcium ion influx promotes the release of azurophilic granules. ii. CDK4/6 phosphorylates and destroys nuclear envelope proteins, while NE is activated by MPO, which in turn promotes an increase in GSDMD expression, thereby destroying the nuclear and plasma membrane structure. iii. Histone citrullination caused by the PAD4 pathway, NE cleavage of histone H1/H4, and increased entropic pressure, leading to chromatin depolymerisation and expansion; iv. Subsequent to the plasma membrane rupturing due to various pathways such as GSDMD and NE, NETs are released in large quantities; their main components are histones, MPO, NE, and DNA networks. Abbreviations: PMA (12-myristoyl-13-acetoxy-phorbol), FcγRII (Fcγ receptor II), Raf-MEK-ERK (Raf/mitogen-activated protein kinase/extracellular signal-regulated kinase pathway), PKC (protein kinase C), ROS (reactive oxygen species), MPO (myeloperoxidase), NE (neutrophil elastase), CDK4/6 (cyclin-dependent kinase 4/6), F-actin (fibrillar actin), PAD4 (peptidylarginine deiminase 4), GSDMD (gasdermin D), NETs (neutrophil extracellular traps).

**Figure 3 ijms-26-02336-f003:**
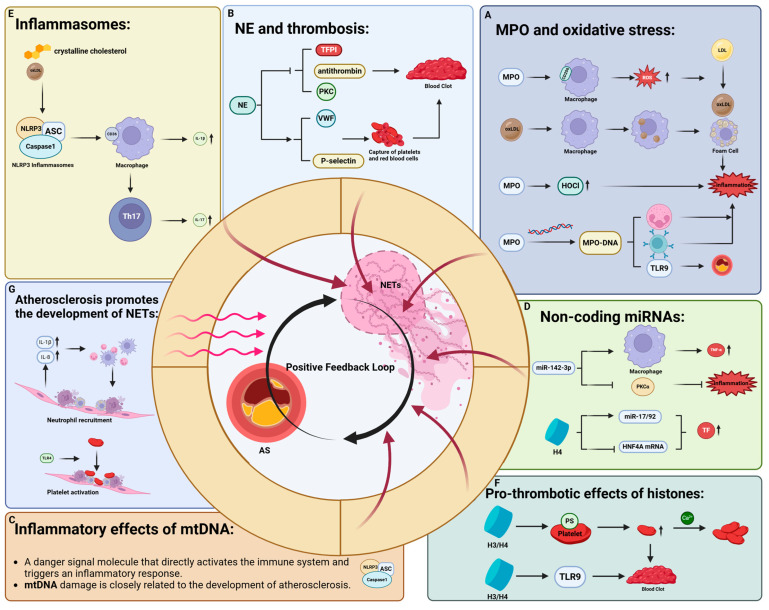
This figure shows the role of NETs in AS and its related mechanisms. Black upward arrow indicates the expression of this substance is upregulated; black arrow indicates the former promotes the upregulation of expression or cell behavior of the latter. The dark red arrow indicates the corresponding parts, which are the main components of NETs and promote the occurrence and development of AS. The diagram illustrates the role of each component of NETs in the development of atherosclerosis: (**A**). MPO can catalyse the formation of oxidised low-density lipoprotein, which promotes the transformation of macrophages into foam cells. It can also activate immune cells by forming MPO-DNA complexes, promote inflammatory responses, and promote the development of atherosclerosis. (**B**). NE promotes thrombosis by degrading TFPI, PKC and antithrombin. (**C**). mtDNA can directly activate the immune system, trigger inflammatory responses, and exacerbate AS by enhancing the inflammatory response and monocyte aggregation; (**D**). miRNAs can activate macrophages, upregulate TNF-α, and trigger the inflammatory response, and histone H4 has also been shown to upregulate the expression of some miRNAs, which in turn upregulates TF expression. It has been observed that miR-142-3p regulates macrophage function by downregulating PKCα, thereby affecting the production of inflammatory mediators. (**E**). Inflammasomes have been shown to promote the production of IL-1β by activating the CD36 receptor on the surface of macrophages, thereby causing an inflammatory response. Macrophages can activate Th17, resulting in increased IL-17 expression. (**F**). Histones H3/H4 bind to phospholipids on the surface of platelets, activating platelets and promoting thrombosis. In addition, they can also activate TLRs and trigger an inflammatory response. (**G**). Immune cell infiltration and platelet activation caused by atherosclerosis also promote the NETosis process. Abbreviations: AS (atherosclerosis), NETs (neutrophil extracellular traps), MPO (myeloperoxidase), NE (neutrophil elastase), PKC (protein kinase C), LDL (low-density lipoprotein), oxLDL (oxidised low-density lipoprotein), ROS (reactive oxygen species), PS (phosphatidylserine), TFPI (tissue factor pathway inhibitor), VWF (von Willebrand factor), TLRs (Toll-Like Receptor), TF (tissue factor), mtDNA (mitochondrial DNA), NLRP3 (NLR Family Pyrin Domain Containing 3), miRNAs (MicroRNAs), TNF-α (Tumour Necrosis Factor-alpha), Th17 (T Helper 17), CD36 (Cluster of Differentiation 36).

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
