# Peer review of "Neutrophil Extracellular Traps in Atherosclerosis: Research Progress"

_ijms, 2025, doi:10.3390/ijms26052336_

Round 1
Reviewer 1 Report
Comments and Suggestions for Authors
I recommend the manuscript "Research Progress of Neutrophil Extracellular Trap Net in Atherosclerosis” to be accepted after major revision done. Therefore, I propose:
1) Point 2.1 contains a lot of general information that does not contribute anything new to the article. Some of this data is not supported by literature sources, e.g. lines 63-66). Additionally, one sentence describes factors that contribute to the formation of NETs, ​​such as ROS, NF-κΒ, cGAS-STING, and the remaining paragraphs describe cellular pathways in which the above factors play a role. The pre-NET phase is extremely important for the pathogenesis of atherosclerosis, which is why this chapter requires specific data.
2) Line 38: Lupus is an autoimmune disease. It is listed separately as if the pathophysiology of the disease were not related to the production of auto-antibodies.
3) Line 42: The authors cite data collected and arranged in a manner of their own choosing, this is not a discussion of the results obtained.
4) Line 139: There is no information about the surface antigens of thrombocytes and/or endothelial cells activating neutrophils.
5) Lines 139-141: The authors write that: "any stimulus can cause neutrophil to actively release NETs". There is no information about physiological and pathological stimulants, especially those related to atherosclerosis. The term “any” is too general.
6) Line 179: The information about 70% ATP consumption must be supported by literature sources.
7) Line 227: What means translated with DeepL.com?
8) Lines 252-256: There is no connection between plasmacytoid dendritic cells and the formation of atherosclerotic plague.
9) Line 342: What exactly are antimicrobial proteins? This information is too general.
10) Lines 348-350: There is no information about which antigens presented on surface of plates, red blood cells and vascular cells NETs bind to.
11) Line 368: Cholesterol and ox-LDL are not antigens, but factors on the basis of which atherosclerotic plaque is created.
12) Figure 1 and 2: All abbreviations should be explained on or below the figure. Additionally, not all elements contained in the figures have been described. Small parts of the figures are not visible.
13) Abbreviations should be explained on first use. It is unacceptable to use abbreviations without explanation.
14) Point 4: Instead of discussion there should be conclusion.
Author Response
Reviewers' comments:
Reviewer #1:
I recommend the manuscript "Research Progress of Neutrophil Extracellular Trap Net in Atherosclerosis” to be accepted after major revision done. Therefore, I propose:
Main points:
- Point 2.1 contains a lot of general information that does not contribute anything new to the article. Some of this data is not supported by literature sources, e.g. lines 63-66). Additionally, one sentence describes factors that contribute to the formation of NETs, ​​such as ROS, NF-κΒ, cGAS-STING, and the remaining paragraphs describe cellular pathways in which the above factors play a role. The pre-NET phase is extremely important for the pathogenesis of atherosclerosis, which is why this chapter requires specific data.
Response:
Thank you for pointing out the issue. Upon reviewing Section 2.1, I realized that I had devoted too much space to describing cellular pathways, which is not aligned with the core focus of this section. I fully agree with your observation, and I appreciate your insight. Regarding the cytokine section, I directly stated that the secretion of pro-inflammatory cytokines plays a key role in the formation of NETs, but I failed to provide relevant literature to support this point. This was an oversight on my part. Therefore, in line with your suggestions, I have supplemented this section with three additional references to better substantiate my argument. I sincerely appreciate your feedback, as it has made this part of the content more rigorous.
Additionally, the excessive description of cellular pathways in Section 2.1 was redundant and did not maintain logical consistency throughout the article. I agree with your recommendation and, following the editor's advice, I have now expanded on the role of ROS and NF-κB in promoting NET formation. I have also simplified most of the cellular pathways that are not directly relevant to the occurrence and development of NETs and atherosclerosis. Based on your final suggestion, I have added information on how the factors contributing to the development of NETs in the pre-NET stage are involved in the pathogenesis of atherosclerosis. Thank you once again for your valuable input, which has greatly enriched this section of the article.
- Line 38: Lupus is an autoimmune disease. It is listed separately as if the pathophysiology of the disease were not related to the production of auto-antibodies.
Response:
Thank you very much for pointing this out. I completely agree with your observation that systemic lupus erythematosus (SLE) is an autoimmune disease, not an inflammatory disease. The intention of separately listing it was to illustrate that if NETs are not cleared in a timely manner, they can accumulate in the body and lead to various diseases. Therefore, SLE was meant to serve as an example, but I realize that my wording was imprecise and led to confusion by classifying SLE as an inflammatory disease.
To address this, I have revised the paragraph to clearly differentiate atherosclerosis, which is driven by inflammatory responses, from autoimmune diseases such as SLE and rheumatoid arthritis. I appreciate your careful attention to this detail, and I believe this revision improves the clarity and accuracy of the manuscript.
- Line 42: The authors cite data collected and arranged in a manner of their own choosing, this is not a discussion of the results obtained.
Response:
Thank you very much for pointing this out. I appreciate your feedback. In my previous version, I organized the process of NETs generation chronologically, focusing on the stages of the process rather than discussing the results. As a result, I realize that this section was not appropriately placed in the Results section.
To address this, I have revised the structure of the manuscript, moved the relevant content out of the Methods section as you suggested, and placed it in the Introduction section to align with the classification method I explained earlier. I hope this adjustment makes the article structure clearer and easier to follow. Thank you again for your valuable suggestion.
- Line 139: There is no information about the surface antigens of thrombocytes and/or endothelial cells activating neutrophils.
Response:
Thank you for your valuable feedback. I completely agree with your comment that I did not include information about the role of thrombocyte and/or endothelial cell surface antigens in activated neutrophils. I realize that this was an oversight on my part and thank you for pointing this out. Based on your suggestion, I have now revised the section to include the thrombocyte and/or endothelial cell surface antigens you mentioned as missing and attached the relevant references.
- Lines 139-141: The authors write that: "any stimulus can cause neutrophil to actively release NETs". There is no information about physiological and pathological stimulants, especially those related to atherosclerosis. The term “any” is too general.
Response:
Thank you very much for pointing this out. The word 'any' in this context was intended to refer to the diverse and complex triggering pathways of NETosis mentioned earlier in the manuscript, specifically the variety of related antigens. But, during the translation process, I mistakenly used 'any,' which led to an unintended shift in meaning.
In response to your feedback, I have revised the wording to more accurately reflect the intended meaning. I also added references to the stimuli associated with atherosclerosis, as discussed earlier in the manuscript, to ensure the accuracy and clarity of the content. Thank you again for your thoughtful observation.
- Line 179: The information about 70% ATP consumption must be supported by literature sources.
Response:
Thank you very much for your insightful comments. I apologize for the oversight during the drafting process, which resulted in the omission of this reference. I have now corrected this and added the appropriate source to ensure proper citation. I appreciate your attention to detail, which has helped improve the accuracy of the manuscript.
- Line 227: What means translated with DeepL.com?
Response:
Thank you for your feedback regarding the inclusion of the phrase 'deepl.com translation' in the manuscript. I sincerely apologize for this oversight. The text was translated using DeepL as part of my drafting process, and I mistakenly left that reference in the document. I understand the importance of adhering to academic standards and ensuring that all references are appropriately cited. I assure you that this does not violate any guidelines related to AI usage. I have now removed this reference from the manuscript and ensured that the content is in compliance with the journal’s requirements. I truly appreciate your understanding and your attention to this matter.
- Lines 252-256: There is no connection between plasmacytoid dendritic cells and the formation of atherosclerotic plague.
Response:
Thank you for your valuable feedback. In this section, I initially focused on the combination of plasmacytoid dendritic cells and MPO-DNA in triggering the IFN-1 response, but I failed to fully elaborate on its connection to the formation of atherosclerotic plaques. I appreciate your observation, and to address this, I have revised the section to provide a more detailed explanation of the specific impact of the IFN-1 response and its role in both the initiation and progression of atherosclerosis. I have also included relevant literature to further substantiate the discussion. Thank you again for your constructive comments, which have significantly enhanced the clarity and depth of this section.
- Line 342: What exactly are antimicrobial proteins? This information is too general.
Response:
Thank you very much for pointing this out. In the previous version of the article, I failed to provide an explanation for the term 'antimicrobial protein' and used it directly, which was not sufficiently rigorous. I sincerely appreciate your feedback, and in response, I have now added a clear definition of antimicrobial proteins and provided a list of those discussed in this review. Your attention to detail has greatly improved the clarity of this section, and I am grateful for your thoughtful suggestion.
- Lines 348-350: There is no information about which antigens presented on surface of plates, red blood cells and vascular cells NETs bind to.
Response:
Thank you very much for your insightful suggestion. Upon reviewing this paragraph, I realized that it does not adequately describe the specific processes involved in the interaction between NETs and blood cells, and it lacks a discussion of surface antigens, which made the section overly general. In response to your feedback, I have rewritten this paragraph to provide a more detailed analysis of the interaction between blood cells and NETs, categorizing the discussion based on different blood cell types and their respective surface antigens.
- Line 368: Cholesterol and ox-LDL are not antigens, but factors on the basis of which atherosclerotic plaque is created.
Response:
Thank you very much for your thoughtful suggestion. I realize that it was incorrect to directly classify cholesterol and ox-LDL as antigens in this paragraph. In light of your feedback, particularly from point 14, I have revisited and revised the discussion section of the article, and I have also added a conclusion section. As a result, this paragraph will be improved and rewritten accordingly.
I sincerely appreciate your correction of the academic errors in this article, which has significantly improved its quality.
- Figure 1 and 2: All abbreviations should be explained on or below the figure. Additionally, not all elements contained in the figures have been described. Small parts of the figures are not visible.
Response:
Thank you very much for your constructive comments. Upon reflection, I realize that my explanation of the figures was not sufficiently clear, and I acknowledge that I had used numerous abbreviations in the original figures without providing adequate explanations. In response to your feedback, I have revised the descriptions to be more specific, provided clearer summaries of the figures, and included explanations for all the abbreviations used. Furthermore, based on your suggestions and the feedback from another editor, I have made substantial revisions to the entire review. As a result, I have redrawn each figure to align with the revised structure of the review, ensuring that the figures now correspond more closely to the text.
Once again, I sincerely appreciate your valuable comments, which have significantly contributed to improving the quality of this review.
- Abbreviations should be explained on first use. It is unacceptable to use abbreviations without explanation.
Response:
Thank you very much for pointing out this issue. We sincerely apologize for the oversight. Although there is a table at the end of the article that explains the meanings of abbreviations, we realized that some abbreviations, which appear for the first time in the text, were not properly explained. In response to your feedback, we have corrected this and ensured that all abbreviations are now clearly defined in the revised version.
- Point 4: Instead of discussion there should be conclusion.
Response:
Thank you very much for your professional revision suggestions. I benefited a lot from the problems you pointed out in the review, especially the omission of accidentally confusing the conclusion with the discussion part in the article. I have made targeted corrections: the discussion part has been reorganized, focusing on the interpretation of the research results and the analysis of the relevance to the literature; the conclusion part has independently supplemented the core innovation points, practical significance and future research directions to ensure a clear logical level. The revised manuscript has been resubmitted. Thank you again for your rigorous guidance. If there are other adjustments that need to be made, please let me know and I will actively cooperate to improve it.

Reviewer 2 Report
Comments and Suggestions for Authors
This is an interesting review article with adequate novelty. However, some points should be addressed.
- The NET abbreviation should be explained in Abstract and Introduction when firstly reported.
- A method section should be added in the Abstract reporting the databases and the keywords used to collect the data of the review article.
- After Introduction section, a separate section entitled as "methods" should be added describing the relevant information of the collection of the data, the use of keywords as well as the use of exclusion and inclusion criteria.
- The Introduction is too small for a review article. More information should be added to describe both the utility of NETs and NETosis, and to report the underlining mechanisms of NETs and NETosis action.
- The scope of the article should be added at the end of the introduction section.
- How the statements in lines 67-83 are connected with NETs? This should describe with more details.
- The Results section is quite complex and not easily readable. The authors should simplify and explain some molecular mechanisms reported in this section. Subheading could contribute to this direction.
- The Discussion section is too small. It seems more as a Conclusion part of the article and not a Discussion part where the authors should scrutinize the above results.
- Overall, this article needs re-organization in order to be more easily readable and understood as well as to have a better organization relevant to review articles.
Author Response
Reviewers' comments:
Reviewer #2:
This is an interesting review article with adequate novelty. However, some points should be addressed.
Main points:
- The NET abbreviation should be explained in Abstract and Introduction when firstly reported.
Response:
Thank you very much for your valuable comments. After review by the editorial team, we deeply realized that there was a problem in the original text that some professional terms were not annotated in time when they first appeared. In response to this omission, we have completed a comprehensive revision in the revised version: for professional abbreviations that appear for the first time, complete explanations are added in the form of footnotes in the text, and the corresponding term explanation comparison table is added at the end of the text (see Appendix 1 for details). This revision further improves the academic expression standards. In subsequent work, we will strictly implement the editorial standards that require annotations for the first mention, and continue to improve the rigor of academic texts. Thank you again for your professional review comments, which are of great significance to our improvement of publishing standards.
- A method section should be added in the Abstract reporting the databases and the keywords used to collect the data of the review article.
Response:
Thank you very much for your insightful comment. Upon reviewing the abstract, I realize that I failed to clearly state the entry point and perspective of the review, which may have left readers without a clear understanding of the article's analytical approach. In response, I have added a brief introduction to the key terms of the article, which outlines the main areas covered and provides a clearer perspective. I sincerely appreciate your feedback, as it has significantly improved the readability and rigor of the article.
- After Introduction section, a separate section entitled as "methods" should be added describing the relevant information of the collection of the data, the use of keywords as well as the use of exclusion and inclusion criteria.
Response:
Thank you for your valuable suggestion. We acknowledge the importance of transparency in data collection. However, as this is a review and not original research, it does not cover original data collection, systematic methods, etc. Therefore, we did not choose to supplement the methods section.
However, we would like to thank you once again for your suggestions. We have also identified a problem in our text, namely that the wrong content was placed in the results section, which is a wrong classification method. Therefore, I have improved this part and described in detail the thinking behind the classification as the background of this review, and finally placed it in the introduction. This ensures clarity while maintaining the usual structure of a review article.
Thank you again for your valuable feedback, which helps us improve the clarity and logical structure of the manuscript.
- The Introduction is too small for a review article. More information should be added to describe both the utility of NETs and NETosis, and to report the underlining mechanisms of NETs and NETosis action.
Response:
Thank you very much for your valuable suggestion. Upon reviewing the introductory section, I realize that it was indeed too brief. To address this, I have expanded the content to include a more detailed explanation of NETs, as well as an introduction to NETosis, in order to make the section more comprehensive and informative.
- The scope of the article should be added at the end of the introduction section.
Response:
Thank you very much for your thoughtful comment. Upon review, I recognize that the introduction indeed lacked a clear statement regarding the scope of the article. In response, we have now specified that the focus of this review is primarily on molecular biology. Additionally, we have included a summary to the introduction to better outline the context in which NETs are involved, thereby making the section more comprehensive. Thank you again for your valuable feedback, which has greatly contributed to improving the structure and completeness of the article.
- How the statements in lines 67-83 are connected with NETs? This should describe with more details.
Response:
Thank you for pointing out the problem. Indeed, the focus of this paragraph is on the description of the cellular pathways of interleukins in the body. However, this part should focus on the role of factors related to the production of pro-NETs in the development of NETs, and its potential impact on the development of atherosclerosis. Therefore, the content and focus of this paragraph are somewhat off. Therefore, I improved part of the content here and added the effects of cytokines on atherosclerosis and NETs to make it more in line with the main theme of this article.
- The Results section is quite complex and not easily readable. The authors should simplify and explain some molecular mechanisms reported in this section. Subheading could contribute to this direction.
Response:
Thank you very much for your insightful suggestion. Upon reviewing the Results section, I recognize that it was indeed cluttered and not well-organized, which made it difficult to read and contributed to redundancy. In response to your feedback, I have restructured the section and introduced subheadings to better organize the content into three distinct parts, making it clearer and easier to read and understand.
- The Discussion section is too small. It seems more as a Conclusion part of the article and not a Discussion part where the authors should scrutinize the above results.
Response:
Thank you very much for your professional and thoughtful suggestions for revision. I have greatly benefited from your comments, particularly regarding the unexpected mixing of the conclusion and the discussion. In response to your feedback, I have made targeted revisions: the discussion section has been reorganized to focus on the interpretation of the research results and an analysis of the relevance of the literature, while the conclusion has been expanded to independently address core innovations, practical significance, and future research directions, ensuring a clear logical structure. The revised manuscript has been resubmitted. I sincerely appreciate your critical guidance, and if there are any further adjustments needed, please do not hesitate to let me know. I will be happy to make any additional improvements.
- Overall, this article needs re-organization in order to be more easily readable and understood as well as to have a better organization relevant to review articles.
Response:
Thank you for pointing out the problems. This review has many unreasonable structural aspects, such as the lack of concepts such as NETosis and atherosclerosis in the introduction. The results section is not well classified, and there is a lack of discussion and an incomplete conclusion. These problems are the main concerns.
Therefore, the structure of the article has been changed to a large extent, including the following aspects: the abstract section, the expansion of the introduction section, the transfer of some content in the results section, the results section is divided into three sections, and further divided using subheadings. An additional conclusions section has been added, and the content of the existing discussion section has been revised and placed in the conclusions section, and the content of the discussion section has been added.
Thank you again for pointing this out, which has enriched the content of this review and improved its readability and comprehensibility.

Round 2
Reviewer 1 Report
Comments and Suggestions for Authors
I accept the authors' explanations and corrections.
Reviewer 2 Report
Comments and Suggestions for Authors
The authors have significantly improved their manuscript.